# Interocular asymmetry of the superonasal retinal nerve fibre layer thickness and blood vessel diameter in healthy subjects

Angelica Ly[1,2], Jennifer Banh[2], Patricia Luu[2], Jessie Huang[1,2], Michael Yapp[1,2], Barbara Zangerl[1,2]*

1 Centre for Eye Health, Sydney, New South Wales, Australia, 2 School of Optometry and Vision Science, University of New South Wales, Sydney, New South Wales, Australia

* b.zangerl@unsw.edu.au

**Data Availability Statement:** All relevant data are within the manuscript and its Supporting Information files.

## Abstract

### Background

Optical coherence tomography is commonly used to measure the retinal nerve fibre layer thickness in both normal and diseased eyes; however, variation among normal eyes is common and may limit the usefulness of the results. The aim of this study was to explore the interocular asymmetries in retinal nerve fibre layer thickness in a group of normal eyes and to investigate the influence of blood vessel diameter on local retinal nerve fibre layer thickness.

### Methods

In this prospective study, retinal nerve fibre layer thickness and blood vessel diameter across 100 healthy participants were measured using two optical coherence tomography instruments. Individuals were categorised into two groups based on the presence or absence of interocular retinal nerve fibre layer thickness asymmetry beyond the 75th percentile of all participants.

### Results

The superonasal sectoral retinal nerve fibre layer thickness was significantly greater in the left eye compared to the right, across all three sectors. Mean blood vessel diameter showed a corresponding difference in thickness at one of the superonasal sectors. Linear regression showed a positive and moderate correlation between blood vessel diameter and focal retinal nerve fibre layer thickness. This trend persisted across both arteries and veins, but veins showed larger variability between left and right eye in participants with marked superonasal retinal nerve fibre layer asymmetry.

### Conclusion

Retinal nerve fibre layer thickness and blood vessel diameter vary significantly between eyes even in healthy individuals. These asymmetries in a normal population should be

 

**Funding:** This work was supported by a National Health and Medical Research Council (NHMRC https://www.nhmrc.gov.au/) grant (#1033224). Guide Dogs NSW/ACT (https://www.guidedogs.com.au/) is a partner in the NHMRC grant and provided support for AL and BZ. The funders had no role in study design, data collection and analysis, decision to publish, or preparation of the manuscript.

**Competing interests:** The authors have declared that no competing interests exist.

taken into consideration when interpreting the retinal nerve fibre layer thickness measurements from optical coherence tomography to assist in distinguishing normal variations from disease.

## Introduction

Optical coherence tomography (OCT) is a non-invasive, diagnostic imaging technology based on the principle of low coherence interferometry that has revolutionised the assessment of ocular disease. In optic neuropathy, it provides an *in vivo* method of quantifying the retinal nerve fibre layer (RNFL) thickness, which may be compared between eyes of the same patient or against a normative database. Glaucoma and other optic neuropathies are often bilateral but asymmetrical. Interocular RNFL thickness asymmetry may therefore provide a useful indication of early disease.[1, 2] However, the thickness and distribution of the RNFL in eyes both with and without disease varies considerably. This physiological variability and other abnormalities present in the healthy optic nerve and non-glaucomatous optic neuropathies also share overlapping features with glaucoma, further complicating the diagnostic process.[3]

To date, there have been several studies describing the effect of age, refractive error, axial length and optic disc size on RNFL thickness.[4] Globally, the impact of blood vessel location and trajectory on RNFL thickness has also described albeit results were inconclusive.[5, 6] Thus, the aim of this study was to investigate interocular asymmetry of the RNFL and its association with blood vessel diameter in healthy subjects. A greater understanding of normal inter-individual variation of the RNFL and associated inter- and intraocular factors will contribute to the understanding and interpretation of OCT in optic nerve disease.

## Methods

### Participants

Participants aged 18 to 84 years who attended the Centre for Eye Health (CFEH), UNSW Sydney, New South Wales, Australia were recruited for this cross-sectional, prospective study between February and May 2017. This study was approved by the Human Research Ethics Advisory panel 'H' of the University of New South Wales, Sydney (Reference number HC 08/2014/36. Patient written consent was obtained for all participants in accordance with the declaration of Helsinki. All participants underwent a comprehensive assessment (including an ocular and medical histories questionnaire, visual acuity, tonometry, perimetry, ocular imaging and stereoscopic optic nerve head assessment) to confirm the absence of any optic nerve pathology that might affect the RNFL. Demographic and clinical data including age, sex and ethnicity, were extracted from the CFEH patient management system (VIP.net, Best Practice Software Pty Ltd, Queensland, Australia).

Exclusion criteria included any structural or perimetric evidence of glaucoma or optic neuropathy in either eye, an intraocular pressure greater than 22 mmHg in either eye at any time, a best-corrected visual acuity worse than 6/12 in either eye or binocularly, amblyopia, strabismus (including microtropia), and a refractive error or spherical equivalent of greater than ±6.00 dioptres (per the consensus threshold of Filtcroft et al.[7]). Participants demonstrating evidence of diabetic retinopathy, previous events such as arteritic or non-arteritic anterior ischaemic optic neuropathy, retinal vessel occlusions, other optic neuropathy, a history of

ocular trauma in either eye, or a systemic history of hypertension or cardiovascular disease were also excluded.

## Data acquisition

In lieu of a "ground truth", peripapillary RNFL thickness measurements were obtained from the Cirrus HD-OCT (Carl Zeiss Meditec, Dublin, California, USA) for classification and the analysis was conducted using independent Spectralis HRA+OCT (Heidelberg Engineering Inc., Heidelberg, Germany) measurements. Subjects with insufficient scan quality i.e. OCT scans showing incorrect segmentation, signal strength less than 7 on the Cirrus or less than 20dB on the Spectralis were ineligible. Using both OCT instruments, RNFL thickness is quantified and reported in a circumpapillary high-resolution fashion both globally and locally, either subdivided into quadrants or sectors (denoted as "clock hours") or represented graphically as the TSNIT profile. Healthy subjects classically show increased thickness superiorly and inferiorly, consistent with what is often described as a "double hump" (Fig 1). As the study was looking at inter-eye symmetry, data from both eyes were included in the analysis.

Based on pilot data and the results of a recent peer-reviewed publication showing significant superonasal interocular RNFL asymmetry,[8] subjects were initially classified using Cirrus OCT sectoral RNFL thickness values between 9 o'clock to 2 o'clock. Subjects were enrolled into the asymmetry group if the difference in RNFL thickness of either corresponding superonasal sectors between the right and left eyes (i.e. one o'clock of the right eye against 11 o'clock in the left eye and 2 o'clock in the right eye versus 10 o'clock in the left eye; Fig 1A) was equal to or greater than the 75th percentile of the average difference of all participants. Participants within the 75th percentile range of the cohort formed the control group.

Blood vessels cutting across the superior hemifield of the OCT scan circle in both eyes of each participant were then located, categorised as either an artery or vein, mapped and measured using the Spectralis SD-OCT glaucoma module premium edition of the Heidelberg Eye Explorer software. The instrument uses a proprietary "anatomic positioning system" to automatically delineate the boundaries of the RNFL along a calculation circle according to fixed landmarks and a line connecting the centre of the fovea and the centre of Bruch's membrane opening (taken as the margins of the optic disc). For the purposes of this study, the RNFL thickness values at each scan location along a 3.5mm diameter scan circle and the corresponding measurement numbers were also extracted using an export function for each patient. Blood vessel location and corresponding RNFL thickness measurements were sorted by sector into six groups at 30° intervals; 0° was used to denote the axis connecting the fovea to Bruch's membrane opening.

Each of blood vessel location, diameter and corresponding RNFL thickness were calculated as follows based on a total of 768 A scan measurements in the calculation circle around the disc:

$$Location \; (degrees) = Measurement \; number/768 \times 360$$

In locating each blood vessel, the software selector bar (Fig 1B) was placed at the beginning (Location A) and end (Location B) of the blood vessel shadow on the Spectralis RNFL scan. The above formula was applied to the measurement number, which corresponds to the blood vessel location using the X-axis scale of the TSNIT curve, beneath the selector bar. Once the measurement number was approximated for the "start" and "end" points on the blood vessel shadow, the corresponding RNFL thickness values were recorded. The midpoint of the blood vessel was calculated in degrees based on the mean of Location A and B, and corresponding RNFL thickness at that location was similarly determined based on the mean of thickness measurements at locations A and B. This procedure allowed the best approximation of the average

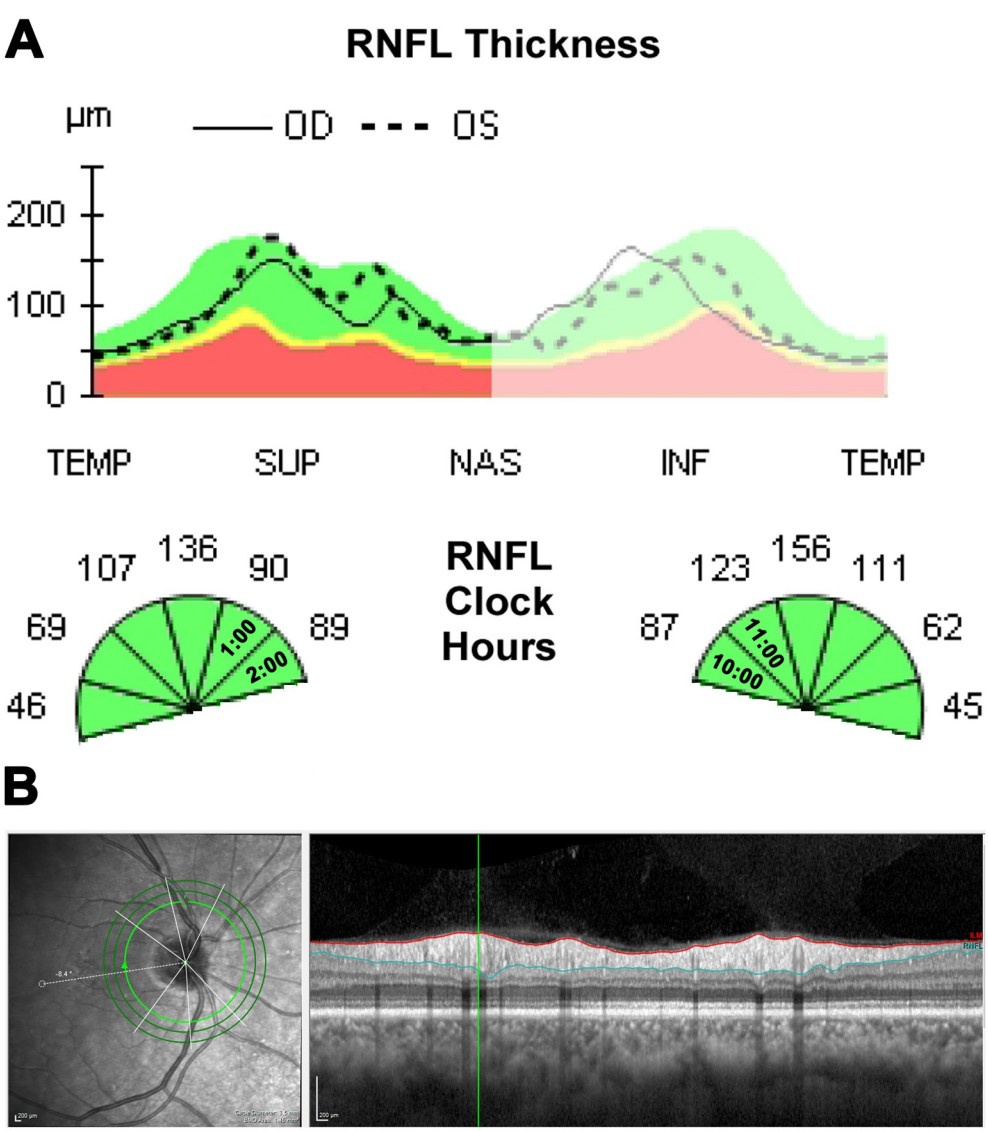

**Fig 1. Study methods.** (A) Subset of the Cirrus OCT retinal nerve fibre layer analysis showing the TSNIT profile and sectoral thickness values. For the study, patients were first categorised into control or asymmetry groups according to corresponding superonasal sectors in both eyes. The difference in thickness (90 versus 123 microns in this example shown) places this case into the asymmetry group. (B) Blood vessel location, thickness and corresponding RNFL thickness were then quantified using the software selector bar (vertical green line appearing on the OCT B-scan) along the innermost 3.5mm diameter calculation circle of the Spectralis OCT.

thickness at the centre of the blood vessel as direct measurements cannot be accurately obtained due to vessel shadowing. Blood vessel diameter was also converted to micrometres using the following formula:

$$Approximate\ vessel\ diameter\ (\mu m) = [\pi \times 3500)] \times (Location\ B\ -\ Location\ A)/360$$

## Statistical methods

Demographic data were summarised using descriptive statistics. Interocular symmetry was calculated as the absolute value of the difference between the values of the right and left eye

and asymmetry was defined as outside of the 75th percentile of the resulting distribution. The differences in mean RNFL thickness and blood vessel diameter between eyes across each sector was tested for statistical significance using an unpaired t-test for each group (control, asymmetry and all participants). Linear regression analysis was used to evaluate the correlation between mean RNFL thickness and blood vessel diameter for arteries and veins. All statistical analyses were performed using GraphPad Prism (Version 7.03; GraphPad Software, La Jolla, California, USA). P values less than 0.05 were considered statistically significant.

## Results

A total of 100 participants (200 eyes), 56 females and 44 males, aged 23–78 years with a mean refractive error of -0.44±1.76 based on worse eye measurements were included in the final analysis (Table 1). The refractive error of all participants fell within range of the normative databases of both OCT devices used in the study. There was no statistically significant difference between the asymmetry and control groups in terms of age, gender or refractive error.

### Superonasal asymmetry in RNFL thickness and blood vessel diameter

RNFL thickness at examined vessel locations was greatest at 61–90 degrees from the horizontal raphe and showed a steady decrease in thickness elsewhere (Fig 2A). Sectoral RNFL thickness was also significantly greater in the left eye compared to the right eye superonasally (Table 2, 91–120˚ and 121–150˚), and reversed nasally (Table 2, 151–180˚). This trend was consistent across corresponding locations in the control group (Fig 2B), although the greater overall thickness in the left eye between 91–150˚ was largely driven by participants in the asymmetry group (Fig 2C).

Overall 1,251 blood vessels were mapped across all eyes, whereby 712 were arteries and 539 were veins. On average, veins were significantly thicker than arteries (109.01±36.37μm vs. 86.04±30.28μm, T-test p<0.001). Notably, there were more arteries than veins in all but the superonasal 91–120˚ sector (which contained 88 arteries and 131 veins). Mean blood vessel diameter did not significantly differ between the two eyes but tended to be greater in the right eye between 61˚ and 120˚, i.e. superiorly, in the asymmetry group (Fig 3). These differences were not appreciable using the mean blood vessel diameter across the total cohort or the control group.

### Blood vessel diameter and mean RNFL thickness

Inter-subject linear regression revealed a statistically significant correlation between blood vessel diameter and mean RNFL thickness at each blood vessel location in both eyes (Fig 4). The

**Table 1. Demographic and ocular characteristics of all participants, asymmetry and control patients, recorded as mean and standard deviation or total count (sex).**
P-value denotes statistical significance comparing the asymmetry against the control group.

|  | All n = 100 | Asymmetry n = 45 | Control n = 55 | P-value |
|---|---|---|---|---|
| Age (years) | 52.91 ± 12.09 | 51.71 ± 13.16 | 53.89 ± 10.46 | 0.37* |
| Sex females/males | 56/44 | 27/18 | 29/26 | 0.47^ |
| Refractive error (D) | -0.44 ± 1.76 | -0.44 ± 1.75 | -0.44 ± 1.80 | 0.95* |
| RNFL OD (μm) | 92.46 ± 8.66 | 93.38 ± 9.39 | 91.71 ± 8.03 | 0.34* |
| RNFL OS (μm) | 92.06 ± 8.55 | 90.76 ± 7.37 | 93.13 ± 9.33 | 0.17* |

D = dioptre;

*unpaired T-test;

^ Chi-square test

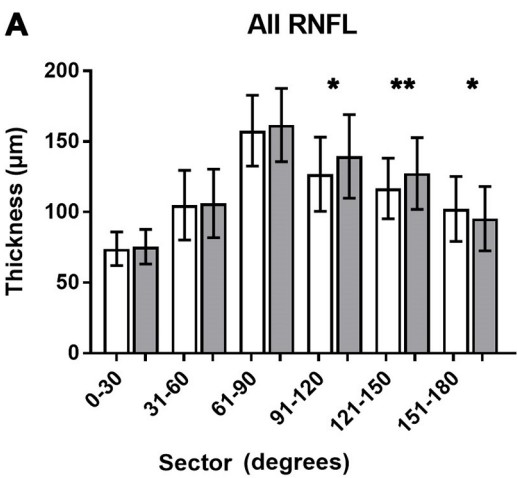

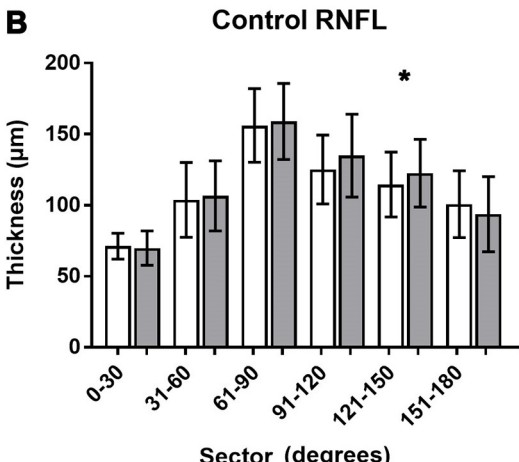

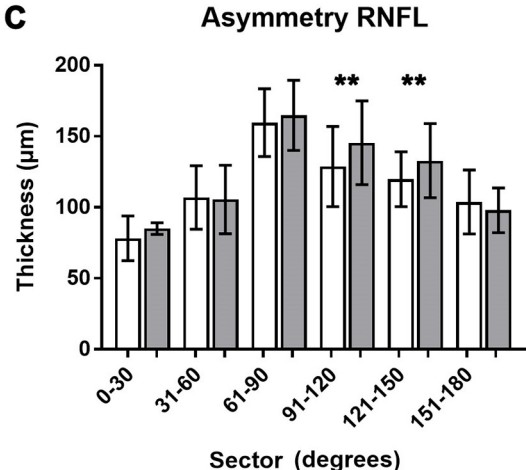

**Fig 2. Comparison of mean RNFL thickness measured at vessel locations between right and left eye.** RNFL measurements at each vessel location were averaged across all participants for each 30° interval and compared between the right (white bars) and left (grey bars) eye. Averages and standard deviations were plotted for all participants (A), those without RNFL asymmetry at the 75[th] percentile (B) and the asymmetry group (C). * = statistically significant difference at $p < 0.05$; ** = statistically significant difference at $p < 0.01$.

**Table 2. RNFL thickness and blood vessel diameter measured across the various sectors (30 degree) for all participants.** Sectoral locations between 0 and 90 represent the superotemporal quadrant, 91–180 the superonasal quadrant. P-value denotes statistical significance between eyes.

| | Right eye | Left eye | Difference | P-value |
|---|---|---|---|---|
| **RNFL thickness (µm)** | | | | |
| 0–30 degrees | 74.0±11.3 (n = 10) | 75.5±11.5 (n = 8) | -1.55 | 0.7900 |
| 31–60 degrees | 104.9±24.3 (n = 40) | 106.2±23.9 (n = 46) | -1.27 | 0.8152 |
| 61–90 degrees | 157.6±25.0 (n = 176) | 161.7±25.8 (n = 172) | -4.04 | 0.1404 |
| 91–120 degrees | 126.8±26.1 (n = 112) | 139.5±29.5 (n = 107) | -12.67 | 0.0456 |
| 121–150 degrees | 116.7±21.5 (n = 162) | 127.4±25.3 (n = 169) | -10.63 | P<0.0001 |
| 151–180 degrees | 102.2±22.9 (n = 121) | 95.1±22.6 (n = 106) | 7.15 | 0.0193 |
| **Mean BV diameter (µm)** | | | | |
| 0–30 degrees | 54.4±11.9 (n = 10) | 57.3±39.0 (n = 8) | -2.86 | 0.8326 |
| 31–60 degrees | 76.6±24.4 (n = 15) | 80.6±19.4 (n = 46) | -4.02 | 0.3980 |
| 61–90 degrees | 116.8±34.1 (n = 176) | 114.5±33.5 (n = 172) | 2.35 | 0.5143 |
| 91–120 degrees | 111.1± 37.4 (n = 112) | 108.8±39.2 (n = 107) | 2.30 | 0.6570 |
| 121–150 degrees | 85.4±28.6 (n = 162) | 90.8±25.3 (n = 169) | -5.47 | 0.0659 |
| 151–180 degrees | 77.3±25.7 (n = 121) | 74.3±24.2 (n = 106) | 2.98 | 0.3712 |

correlation was positive, moderate in size and consistent between both eyes (slope 0.48 and 0.47 at $R^2$ = 0.24 and 0.23, p<0.0001 in the right and left eye, respectively) in the control group. This correlation differed significantly in the asymmetry group (F [DFn, Dfd] = 4.8 [2,547], p = 0.008), whereby the right eye was more strongly impacted by vessel diameter (Fig 3C; slope 0.58 and 0.44 at $R^2$ = 0.23 and 0.23, p<0.0001 in the right and left eye respectively). Analysed by vessel type, linear regression models between blood vessel diameter and mean RNFL thickness differed significantly between arteries and veins (F [DFn, Dfd] = 81.32 [2,1247], p<0.0001), driven by the overall larger diameter of veins (100.2 ± 43.0 median and ICR) compared to arteries (85.9± 28.61 median and ICR).

Regression analysis investigating the correlation between blood vessel diameter and RNFL thickness was subsequently investigated for data grouped by sector under consideration of vessel type, eye laterality, and asymmetry classification (Table 3). Pairwise comparisons between groups was not significant for arteries. For veins, the regression line describing the relationship for the left eye of the asymmetry group differed significantly from the other three groups (F [DFn, Dfd] = 2.972 [6,524], p = 0.0073).

Regression analysis further revealed significant correlations between mean blood vessel diameter and RNFL thickness at corresponding locations when averaged across sectors for either vessel type (Fig 5). Correlation was best for arteries ($R^2$ = 0.96) than for veins ($R^2$ = 0.88 and 0.89 for right and left eye, respectively). For both eyes, the thickness of veins in the supero-temporal area (31–60˚) were relatively smaller compared to all other measurements (Fig 5, black box). Notably, the 91–120˚ superonasal sector deviated towards larger than average veins with regard to the respective RNFL thickness compared to all other sectors in the right eye (Fig 5, black arrow). In comparison, the veins in the left eye of the asymmetry group (black circles) were generally associated with a larger RNFL thickness than those of similar size in the other groups (blue circles). As such, differences in vein diameter in the left eye in some individuals appeared to be correlated with the observed pattern of asymmetry.

## Discussion

The RNFL is comprised of ganglion cell axons projecting toward the optic nerve head. Given the overlapping appearance between normal and pathological optic discs and the importance

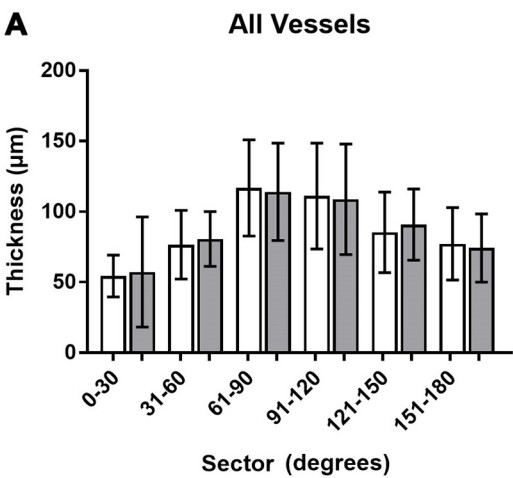

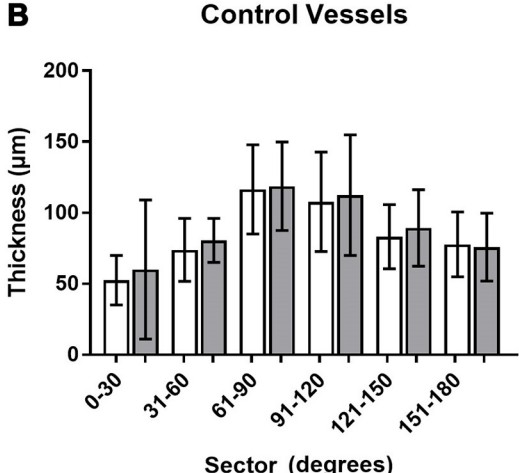

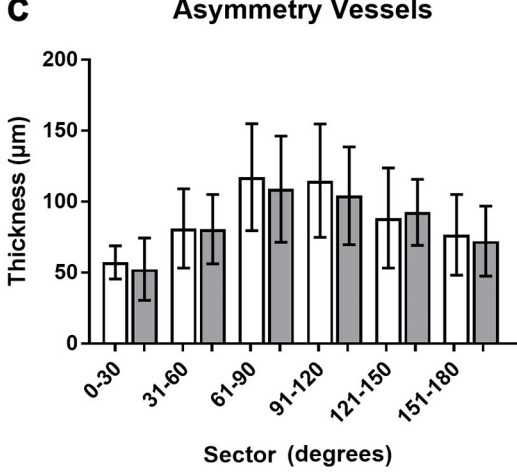

**Fig 3. Comparison of mean blood vessel diameters between right and left eye.** Blood vessel diameters were averaged across all participants for each 30˚ interval and compared between the right (white bars) and left (grey bars) eye. Averages and standard deviations were plotted for all participants (A), those without RNFL asymmetry at the 75th percentile (B) and the asymmetry group (C).

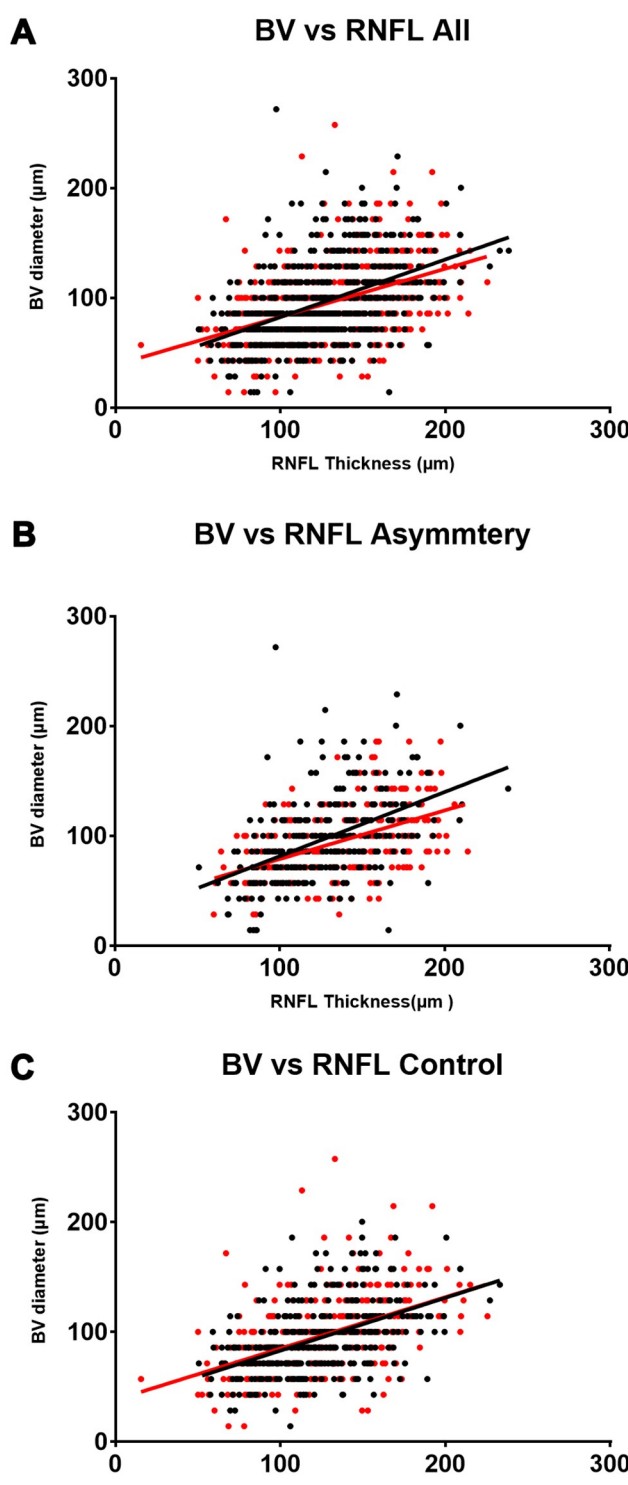

**Fig 4. Scatterplots illustrating the individual relationship between blood vessel diameter and mean RNFL thickness per eye.** The correlations between individual blood vessel diameter measurements and corresponding RNFL thickness is weak ($R^2$ between 0.2153 and 0.2419), but highly significant ($p < 0.0001$). The slight differences in slopes between the right (black) and left (red) eye observed with all participants (A) is highly driven by the divergence in the asymmetry group (B), while the relationship is near identical in the control group (C).

**Table 3. Regression analysis correlating RNFL thickness and blood vessel diameter at vessel locations on average and for each sector (30 degree) by eye and asymmetry classification.** For each group, number of vessel observations (n) and the resulting regression equation (slope and y-intercept) are provided. Based on F-test statistics, arteries and veins always resulted in significantly different correlations. Most pairwise comparisons of regression equations describing the relationship between RNFL thickness and vessel diameter were statistically inseparable, but the correlation obtained for veins in the left eye of the asymmetry group (*) was significantly different from the respective control group as well as the right eye of the asymmetry group.

| | Right Eye | | Left Eye | |
|---|---|---|---|---|
| | **n** | **Regression** | **n** | **Regression** |
| **Arteries** | | | | |
| Control | 194 | 0.62x + 6.96 | 189 | 0.52x + 22.77 |
| Asymmetry | 164 | 0.64x + 8.68 | 150 | 0.57x + 11.01 |
| **Veins** | | | | |
| Control | 153 | 0.78x + 9.25 | 148 | 0.83x + 1.73 |
| Asymmetry | 110 | 0.62x + 27.39 | 121 | 0.71x + 3.02* |

**Fig 5. Correlation between mean blood vessel diameter and corresponding RNFL thickness averaged for sector per vessel type.** A strong positive correlation was observed between arteries (red, p = 0.0005) and veins (blue, p = 0.0057) respectively and corresponding RNFL thickness. The data demonstrate a direct relationship for increasing thickness, which was lowest in the temporal sector (0–30˚) and highest in the superotemporal sector (60–90˚). Linear regression for veins in the left eye of the asymmetry group (depicted as black circles for sectors located between 31˚ and 150˚) differed significantly from all other analyses performed for veins (Table 3). While all veins located in the lower superonasal area (31–60˚) were associated with larger than expected RNFL thickness (black box), a notable deviation in the opposite direction was noted for the higher superonasal sector (90–120˚) for all veins (black arrow) when compared to those of the left eye or the asymmetry group or the association identified for arteries.

of early detection, structural interocular asymmetry is often used as an early indicator of disease.[1, 9] Values falling outside of the normal limits may be regarded as a sign of early disease and this approach has been commonly extended to other variables, including intraocular pressure, cup to disc ratio of the optic nerve head and visual field indices. These results describe a common though under-recognised finding of interocular asymmetry in the superonasal RNFL and its moderate correlation with blood vessel diameter among a group of normal subjects. Superonasal patterns of RNFL asymmetry warrant particular consideration in pigmentary glaucoma, mitochondrial optic neuropathies (Leber's hereditary optic neuropathy and dominant optic atrophy), retrograde degeneration and high-myopia associated optic neuropathy.

Interocular asymmetries in the RNFL using spectral domain OCT of healthy subjects have been described previously in the literature.[1, 2, 4, 10] Similar to those previous reports that showed a mean interocular variation in global RNFL thickness between -0.9 and 3.58μm, the present study found a global RNFL thickness asymmetry of 0.4μm, although ranging between -12.86 and 4.17 between individual clock hours of the superior RNFL (thickness of right minus left eye). In normal subjects, further supported by this study, this asymmetry varies by location and is larger when considering corresponding quadrants or sectors between both eyes.[9] Using 2.5$^{th}$ and 97.5$^{th}$ percentiles and data from 617 subjects, Hwang et al. reported a normal interocular difference in global and quadrant RNFL thickness of 9.5 and 23μm, respectively. They also found a distinctly thinner superior RNFL in the right eye compared to the left. Further consistent with our results, Mwanza et al.[10] and others[1, 9] specifically note this asymmetry to be of greatest magnitude superonasally reporting a mean difference of 11.02μm between the one and eleven o'clock sectors of the right and left eyes,[10] which is mirrored by an average difference of 12.86 μm in the current data. More importantly, this difference increases to an average of 21.13μm in individuals identified with a marked asymmetry, while it reduced to 6.09μm in the control cohort.

We also demonstrated a positive correlation between blood vessel diameter and mean RNFL thickness among superior circumpapillary locations on the scan circle. The hypothesised relationship between blood vessel diameter and RNFL thickness is not new and has been posited previously based on the spatial concordance in rim and RNFL thickness and the observation of larger juxtapapillary retinal vessels in the same location supero- or inferotemporally. The finding that a thicker RNFL relates specifically and in part to greater blood vessel diameter, especially in arteries, has been described previously by Hood et al.[5] who observed the co-localisation of the RNFL maxima either on or just adjacent of the location of the major blood vessels. The authors describe two reasons for the correspondence: 1) direct contribution of the blood vessels to the OCT measured RNFL thickness, and 2) tendency of the arcuate fibres to coincide in location with the major temporal blood vessels.[5]

Thus, although the shape of an individual's OCT measured RNFL profile is well known to relate directly to the location of the superotemporal and inferotemporal major veins and arteries,[6] our study is the first to comprehensively relate blood vessel diameter to RNFL thickness as measured using 1,251 data points from SD-OCT, across a range of both major and minor peripapillary retinal blood vessels in a large cohort of normal eyes. Organ pairs are not perfectly symmetrical[10] and although there was no overall significant difference in mean blood vessel diameter between the left and right eye, veins were larger in the right than the left in individuals with marked asymmetry of the superonasal RNFL thickness. Furthermore, consistent with other work,[11] the diameter of arteries generally correlated better with RNFL thickness than vein diameter, whereby arteries of the same blood vessel diameter than corresponding veins were also associated with larger RNFL thickness (Fig 5). By measuring the range of individual vessels across the whole superior hemifield and the RNFL thickness at corresponding locations, we were able to convincingly show that blood vessel diameter accounted

for up to 24% of the variability in focal RNFL thickness (Fig 4). This exceeds the correlations described in past studies,[1, 9–11] which were based on global or sectoral RNFL thickness values, and other parameters, including age, axial length, refractive error, intraocular pressure, cup to disc ratio, disc and rim area (Pearson correlation coefficient ranging from 0.04 to 0.47), and aligns closely with more recent work by Pereira et al.[12, 13] By relating the RNFL TSNIT profile with a retinal vessel density profile, the latter group found that up to 24% of the interindividual variance of the circumpapillary RNFL distribution may be explained by the distribution of retinal vessels around the optic nerve head. Taken together, these results also demonstrate a significantly thicker superonasal RNFL profile in the left eye (compared to the right) and a corresponding significant difference in the correlation between RNFL thickness and vein diameter, which represents a significant expansion on prior knowledge.

Other possible causes of interocular RNFL asymmetry by sector, not described in this study, include optic disc size asymmetry, refractive error asymmetry, the position of the RNFL scan circle, test-retest variability and age.[9, 10, 14] The first two were controlled for through careful patient selection. The Spectralis OCT also circumvents the problem of scan placement by automatically identifying the fovea and the Bruch's membrane opening demarcated optic nerve head. In the present study, subjects were pre-selected based on an asymmetry observed using Cirrus OCT and measurements made on the Spectralis. Although this dual imaging assessment was designed as a strength, it may also have contributed to the selection bias. Other instrument results with different hardware or software components were also not considered. Another limitation of this work is that it did not include a disease cohort and thus fails to address the broader, arguably more clinically relevant, issue of pathological interocular asymmetries, such as in glaucoma, although the topic has been the focus of other work.[1, 15] Finally, only one RNFL scan per eye was considered in the analysis. Thus, effects due to test-retest variability or scan order could not be exclusively ruled out; however, this issue has been similarly addressed by past studies elsewhere that have shown little to no effect.[9, 16]

In conclusion, the present study showed significant superonasal, interocular RNFL asymmetries in a normal population. These findings were shown to correlate with vein diameter and should be taken into consideration when interpreting the RNFL thickness measurements from OCT.

## Supporting information

**S1 Dataset. Dataset.**
(XLSX)

## Acknowledgments

The authors thank Vivien Cheung for her involvement in preliminary data collection and analysis.

## Author Contributions

**Conceptualization:** Jessie Huang, Michael Yapp, Barbara Zangerl.

**Data curation:** Jennifer Banh, Patricia Luu, Jessie Huang, Barbara Zangerl.

**Formal analysis:** Jennifer Banh, Patricia Luu, Jessie Huang, Barbara Zangerl.

**Funding acquisition:** Barbara Zangerl.

**Investigation:** Barbara Zangerl.

**Methodology:** Jennifer Banh, Patricia Luu, Michael Yapp, Barbara Zangerl.

**Project administration:** Jessie Huang, Barbara Zangerl.

**Supervision:** Jessie Huang, Michael Yapp, Barbara Zangerl.

**Visualization:** Angelica Ly, Barbara Zangerl.

**Writing – original draft:** Angelica Ly.

**Writing – review & editing:** Angelica Ly.

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
