## [Decision Letter · Decision Letter 0]

7 Oct 2019

PONE-D-19-23453

Interocular asymmetry of the superonasal retinal nerve fibre layer thickness and blood vessel diameter in healthy subjects

PLOS ONE

Dear Dr Zangerl,

Thank you for submitting your manuscript to PLOS ONE. After careful consideration, we feel that it has merit but does not fully meet PLOS ONE’s publication criteria as it currently stands. Therefore, we invite you to submit a revised version of the manuscript that addresses the points raised during the review process.

Please analyze your results using multivatiate regression. Please address the reviewers concerns about the lack of novelty of this research: "The correlation between RNFL thickness and blood vessel has been established for decades, so that revisiting the topic with focus on a single sector seems far-fetched, and obviously not a new finding."

We would appreciate receiving your revised manuscript by Nov 21 2019 11:59PM. To enhance the reproducibility of your results, we recommend that if applicable you deposit your laboratory protocols in protocols.io, where a protocol can be assigned its own identifier (DOI) such that it can be cited independently in the future. For instructions see: http://journals.plos.org/plosone/s/submission-guidelines#loc-laboratory-protocols

We look forward to receiving your revised manuscript.

Kind regards,

Alfred S Lewin, Ph.D.

Academic Editor

**PLOS ONE**

**Journal Requirements:**

2. We note that one or more of the authors are employed by a commercial company: Specsavers.

**Comments to the Author**

1. Is the manuscript technically sound, and do the data support the conclusions?

Reviewer #1: Yes

Reviewer #2: Yes

2. Has the statistical analysis been performed appropriately and rigorously? 

Reviewer #1: Yes

Reviewer #2: No

3. Have the authors made all data underlying the findings in their manuscript fully available?

Reviewer #1: Yes

Reviewer #2: Yes

4. Is the manuscript presented in an intelligible fashion and written in standard English?

Reviewer #1: Yes

Reviewer #2: Yes

5. Review Comments to the Author

Reviewer #1: In methodology authors have described -+6D as cut off which is slightly more when we talk about normative data but in results the spherical equivalent is very less which is acceptable.

Age related changes in RNFL can exist and its known fact that every decade there will be some axonal loss which is acceptable for that age.So physiological changes can vary results.

Visual acuity criteria is not very stringent.The authors have mentioned limit as 6/12 which even amblyopes ,mictropic cases can have inspite of normal fundus and RNFL AND Angio parameters will vary from normal in such cases.

This current study has several limitations.Currently, different OCTA systems built by different manufacturers are used worldwide and due to different hardware and software of visualization of vasculature and quantification can vary if different angioOCT are used.

Reviewer #2: In this manuscript, Ly et al assessed the interocular asymmetry of RNFL in normal eyes and investigated whether it is affected by blood vessel diameter at the same location. They reported that left eyes has significantly thicker RNFL than right eyes and that there was RNFL thickness in the superonasal sectors correlated positively with blood vessel diameter.

The paper is clearly and well written and enjoyable to read. Here are a few comments:

1) It is not clear why from the get go the authors chose to focus on the superonasal RNFL. The rationale of the choice was not provided.

2) Relative to remark #1, RNFL in the nasal sector has not proved to be of great value in the diagnosis of certain diseases, i.e glaucoma.

3) The correlation between RNFL thickness and blood vessel has been established for decades, so that revisiting the topic with focus on a single sector seems far-fetched, and obviously not a new finding.

4) From the statistical standpoint, it would have been interesting to run a multivariate regression analysis (with age, axial length, IOP, sex, disc area....) to see the real contribution of blood vessels to RNFL thickness.

6. PLOS authors have the option to publish the peer review history of their article (what does this mean?). If published, this will include your full peer review and any attached files.

Reviewer #1: Yes: NEELAM PAWAR

Reviewer #2: No

---

## [Author Response · Author response to Decision Letter 0]

20 Nov 2019

Response to the Editor

Comment 1

Response

• We thank the editorial team for their instructions.

• The manuscript meets PLOS ONE’s style requirements.

• The regular text used for the equations on line 127 and 140 have both been updated using Equation Tools.

Comment 2

We note that one or more of the authors are employed by a commercial company: Specsavers.

Response

• The author’s affiliations have been updated. 

• We no longer have any commercial company affiliations.

 

Comment 3

Please include captions for your Supporting Information files at the end of your manuscript, and update any in-text citations to match accordingly. Please see our Supporting Information guidelines for more information: http://journals.plos.org/plosone/s/supporting-information.

Response

• A caption for the supporting information file has now been included at the end of the manuscript (lines 396-397, page 19 of the track changes copy).

Response to Reviewer #1

Comment 4

In methodology authors have described -+6D as cut off which is slightly more when we talk about normative data but in results the spherical equivalent is very less which is acceptable.

Response

• Thank you for the feedback. We agree with the reviewer that many normative data studies adopt a spherical refraction equivalent within ±5D but chose to adopt a ±6D cut off given the consensus definition of pathological myopia described by Filtcroft et al. (IOVS 2019; 60: M20-30). This detail has now been added to the methods of the manuscript: “Exclusion criteria included… a refractive error or spherical equivalent of greater than ±6.00 dioptres (per the consensus threshold of Filtcroft et al.[7]).” (lines 74-78 of page 4)

• The refractive range described as an exclusion criteria was quite broad; however, as the reviewer noted, the spherical equivalent refraction of all subjects also fell comfortably within the normative databases of the two instruments used in this study (Cirrus and Spectralis, which include refractive errors ranging from -12 to +8D and -7 to +5D, respectively). The manuscript now reads: “The refractive error of all participants fell within range of the normative databases of both OCT devices used in the study.” (lines 156-157 of page 8)

• Of the 100 subjects enrolled, the maximum spherical equivalent was +2.5D, the minimum was -5.25D.

Comment 5

Age related changes in RNFL can exist and its known fact that every decade there will be some axonal loss which is acceptable for that age. So physiological changes can vary results.

Response

• We agree that RNFL thickness changes with age are a possible confounding variable.

• In response to this comment, we correlated the baseline Cirrus HD-OCT interocular RNFL thickness asymmetry values (between corresponding superonasal RNFL sectors, 1 o’clock in the right eye against 11 o’clock in the left eye and 2 o’clock in the right eye against 10 o’clock in the left eye) and found conflicting results – R2 = 0.04882, P= 0.0272* and R2 = 0.0001526, P = 0.9029 (NS). Thus, and given that the number of enrolled subjects per decade of age was not prospectively standardized, we have now acknowledged this in the text as a study limitation: “Other possible causes of interocular RNFL asymmetry by sector, not described in this study, include optic disc size asymmetry, refractive error asymmetry, the position of the RNFL scan circle, test-retest variability and age.” (lines 327-329, page 16). 

• Fortunately, there was no statistically significant difference in age (p-value of 0.37, reported in table 1) between the control and asymmetry study cohorts (lines 157-158, page 8).

Comment 6

Visual acuity criteria is not very stringent. The authors have mentioned limit as 6/12 which even amblyopes,mictropic cases can have inspite of normal fundus and RNFL AND Angio parameters will vary from normal in such cases.

Response

• No subjects had amblyopia or microtropia and this detail is now included in the updated manuscript: “Exclusion criteria included any structural or perimetric evidence of glaucoma or optic neuropathy in either eye, an intraocular pressure greater than 22 mmHg in either eye at any time, a best-corrected visual acuity worse than 6/12 in either eye or binocularly, amblyopia, strabismus (including microtropia), and a refractive error or spherical equivalent of greater than ±6.00 dioptres.” (lines 76-77, page 4)

Comment 7

This current study has several limitations. Currently, different OCTA systems built by different manufacturers are used worldwide and due to different hardware and software of visualization of vasculature and quantification can vary if different angioOCT are used.

Response

• This limitation is now stated explicitly in the text: “Other instrument results with different hardware or software components were also not considered.” (lines 334-335, page 16)

Response to Reviewer #2

Comment 8

It is not clear why from the get go the authors chose to focus on the superonasal RNFL. The rationale of the choice was not provided.

Response

• Rationale for the choice has been updated in the manuscript as follows: “Based on pilot data and the results of a recent peer-reviewed publication showing significant superonasal interocular RNFL asymmetry,[7] subjects were initially classified using Cirrus OCT sectoral RNFL thickness values between 9 o’clock to 2 o’clock. Subjects were enrolled into the asymmetry group if the difference in RNFL thickness of either corresponding superonasal sectors between the right and left eyes (i.e. one o’clock of the right eye against 11 o’clock in the left eye and 2 o’clock in the right eye versus 10 o’clock in the left eye; Fig 1A) was equal to or greater than the 75th percentile of the average difference of all participants.” (lines 104-110, page 5)

Comment 9

Relative to remark #1, RNFL in the nasal sector has not proved to be of great value in the diagnosis of certain diseases, i.e glaucoma.

Response

• We agree with the reviewer that nasal sector RNFL thickness is of limited value in most subtypes of glaucoma; however, the nasal and superonasal RNFL thickness has been shown to be especially relevant in:

o Pigmentary glaucoma (Baniasadi N, J Glaucoma 2016; 25(10): 865-872) 

o Mitochondrial optic neuropathies, including Leber’s hereditary optic neuropathy and Dominant optic atrophy (Asanad S et al. Curr Eye Res 2019; 44(6):638-644) 

o Retrograde degeneration (Zangerl B et al. CXO, 2017; 100(3): 214-226), as well as 

o Young myopic glaucomatous appearing patients with different optic disc tilt direction (Lee et al. J Glaucoma 2017; 26: 144-152)

• A sentence to this effect has now been included in the discussion: “Superonasal patterns of RNFL asymmetry warrant particular consideration in pigmentary glaucoma, mitochondrial optic neuropathies, retrograde degeneration and high-myopia associated optic neuropathy.” (lines 269-272, page 14)

Comment 10

The correlation between RNFL thickness and blood vessel has been established for decades, so that revisiting the topic with focus on a single sector seems far-fetched, and obviously not a new finding.

Response

• We thank the reviewer for their comment. We agree that RNFL thickness and blood vessel diameter has been previously established and state this explicitly: “The hypothesised relationship between blood vessel diameter and RNFL thickness is not new and has been posited previously based on the spatial concordance in rim and RNFL thickness and the observation of larger juxtapapillary retinal vessels in the same location supero- or inferotemporally. The finding that a thicker RNFL relates specifically and in part to greater blood vessel diameter, especially in arteries, has been described previously by Hood et al.[5] who observed the co-localisation of the RNFL maxima either on or just adjacent of the location of the major blood vessels. The authors describe two reasons for the correspondence: 1) direct contribution of the blood vessels to the OCT measured RNFL thickness, and 2) tendency of the arcuate fibres to coincide in location with the major temporal blood vessels.[5]” (lines 291-300, page 15)

• However, we then go on to highlight the value of the work: “our study is the first to comprehensively relate blood vessel diameter to RNFL thickness as measured using 1,251 data points from SD-OCT, across a range of both major and minor peripapillary retinal blood vessels in a large cohort of normal eyes.” (lines 303-306, page 15).

• The manuscript has also been updated to highlight the unique finding that: “Taken together, these results also demonstrate a significantly thicker superonasal RNFL profile in the left eye (compared to the right) and a corresponding significant difference in the correlation between RNFL thickness and vein diameter, which represents a significant expansion on prior knowledge” (lines 322-325, page 16).

Comment 11

From the statistical standpoint, it would have been interesting to run a multivariate regression analysis (with age, axial length, IOP, sex, disc area....) to see the real contribution of blood vessels to RNFL thickness.

Response

• We thank the reviewer for their interest and request that they kindly consider that the aim of this study was to explore the interocular asymmetry in RNFL thickness in a group of normal eyes. A multivariate regression analysis using interocular RNFL asymmetry as the outcome variable is consequently inappropriate for two reasons: 1) that age, axial length, IOP, sex and disc area are strongly correlated between both eyes, and 2) that multiple blood vessel thickness and RNFL thickness measures were performed per eye.

• The real contribution of the blood vessels to global RNFL thickness relative to these other parameters has been reported previously by Pereira et al. (IOVS, 2015; 56(9): 5290-5298) and the reader is directed to this work in the discussion: “By measuring the range of individual vessels across the whole superior hemifield and the RNFL thickness at corresponding locations, we were able to convincingly show that blood vessel diameter accounted for up to 24% of the variability in focal RNFL thickness (Fig 3). This exceeds the correlations described in past studies,[1, 7-9] which were based on global or sectoral RNFL thickness values, and other parameters, including age, axial length, refractive error, intraocular pressure, cup to disc ratio, disc and rim area (Pearson correlation coefficient ranging from 0.04 to 0.47), and aligns closely with more recent work by Pereira et al.[10, 11] By relating the RNFL TSNIT profile with a retinal vessel density profile, the latter group found that up to 24% of the interindividual variance of the circumpapillary RNFL distribution may be explained by the distribution of retinal vessels around the optic nerve head.” (lines 312-322, page 16).

---

## [Editor Report · Decision Letter 1]

6 Dec 2019

Interocular asymmetry of the superonasal retinal nerve fibre layer thickness and blood vessel diameter in healthy subjects

PONE-D-19-23453R1

Dear Dr. Zangerl,

We are pleased to inform you that your manuscript has been judged scientifically suitable for publication and will be formally accepted for publication once it complies with all outstanding technical requirements. Thank you for your thorough responses to the editorial requests and the concerns of the reviewers.

With kind regards,

Alfred S Lewin, Ph.D.

Section Editor

PLOS ONE
---

## [Editor Report · Acceptance letter]

12 Dec 2019

PONE-D-19-23453R1 

Interocular asymmetry of the superonasal retinal nerve fibre layer thickness and blood vessel diameter in healthy subjects 

Dear Dr. Zangerl:

I am pleased to inform you that your manuscript has been deemed suitable for publication in PLOS ONE. Congratulations! Your manuscript is now with our production department. 

With kind regards,

on behalf of

Dr. Alfred S Lewin 

Section Editor

PLOS ONE